# Denovo Production of Resveratrol by Engineered Rice Wine Strain *Saccharomyces cerevisiae* HJ08 and Its Application in Rice Wine Brewing

**DOI:** 10.3390/jof10080513

**Published:** 2024-07-23

**Authors:** Huihui An, Guangpeng Li, Zhihan Yang, Meng Xiong, Na Wang, Xitao Cao, Aiqun Yu

**Affiliations:** 1School of Biotechnology, Jiangsu University of Science and Technology, Zhenjiang 212100, China; 2State Key Laboratory of Food Nutrition and Safety, Key Laboratory of Industrial Fermentation Microbiology of the Ministry of Education, Tianjin Key Laboratory of Industrial Microbiology, College of Biotechnology, Tianjin University of Science and Technology, No. 29 the 13th Street TEDA, Tianjin 300457, China

**Keywords:** resveratrol, *Saccharomyces cerevisiae*, rice wine

## Abstract

Resveratrol is a plant-derived polyphenolic compound with numerous biological activities and health-promoting properties. Rice wine is a popular traditional alcoholic beverage made from fermented rice grains, and widely consumed in Asia. To develop resveratrol-enriched rice wine, a heterologous resveratrol biosynthesis pathway was established by integrating the 4-coumaroyl-CoA ligase (*Pc4CL*) and the stilbene synthase (*VvSTS*) from *Petroselinum crispum* and *Vitis vinifera* at the δ locus sites of industrial rice wine strains *Saccharomyces cerevisiae* HJ. The resulting *S. cerevisiae* HJ01 produced a level of 0.6 ± 0.01 mg/L resveratrol. Next, the resveratrol production was increased 16.25-fold through employing the fused protein *Pc4CL*::*VvSTS* with a rigidly linked peptide (TPTP, EAAAK). Then, the strains were further modified by removing feedback inhibition of tyrosine through point mutation of *ARO4* and *ARO7*, which integrated at the rDNA region of strain HJ03, and generated strain HJ06, HJ07, and HJ08. Subsequently, the highest resveratrol titer (34.22 ± 3.62 mg/L) was obtained by optimizing fermentation time and precursor addition amount. Finally, resveratrol content of rice wine fermented with strain HJ08 was 2.04 ± 0.08 mg/L and 1.45 ± 0.06 mg/L with or without the addition of 400 mg/L tyrosine after 7 days fermentation.

## 1. Introduction

Recently, with the development of modern society, more and more people have begun to pursue nutritious, healthy, and functional foods. Plant polyphenols, as a class of important bioactive substances, have positive effects on human health and are important ingredients for preparation of functional foods [1]. Resveratrol (3,5,4′-trihydroxy-trans-stilbene), a non-flavonoid polyphenolic compound, belongs to the stilbene class and is widely distributed in some plant species and their derived food products, such as grapes, *Polygonum cuspidatum*, berry varieties, peanuts, and other plants [2,3]. In recent years, numerous in vivo and in vitro studies have revealed that resveratrol possesses diverse biological activities and health-promoting properties, including antioxidant, anti-inflammatory, anti-cancer, anti-aging, cardiovascular protection, neuroprotective, and blood pressure lowering properties [4,5,6,7]. Due to its wide range of biological properties, resveratrol has been widely used as a raw material in different industries, including pharmaceutics, food, and cosmetics [3,8,9]. Worldwide attention has been focused on it. To date, microbial biosynthesis of resveratrol, using metabolic engineering and synthetic biology, provides a promising and sustainable approach because it would provide low-cost cultures, environmental compatibility, and higher productivity compared with plant-based extraction or chemical synthesis processes [10].

In plants, the biosynthetic pathway of resveratrol could be achieved through the shikimate and aromatic amino acid via L-phenyalanine (L-Phe) or L-tyrosine (L-Tyr) as precursors [11]. As shown in Figure 1, there are two branches. Firstly, phenylalanine ammonia lyase (PAL) catalyzes the conversion of L-phe to form cinnamic acid, which is further processed to generate *p*-coumaric acid (*p*-CA) using cinnamate-4-hydroxylase (C4H). Alternatively, tyrosine ammonia lyase can directly use L-tyrosine as a substrate to generate *p*-CA. Next, *p*-coumaroyl coenzyme A ligase (*4CL*) catalyzes the conversion of *p*-CA to form *p*-coumaroyl-CoA. Finally, resveratrol is biosynthesized by the condensation of three units of malonyl-CoA with one molecule of *p*-coumaroyl-CoA by stilbene synthase (*STS*).

Resveratrol has been successfully biosynthesized through heterologous expression in several microorganisms, such as *Escherichia coli*, *Saccharomyces cerevisiae*, *Rhodotorula toruloides*, and *Yarrowia lipolytica* [12,13,14]. The most recent study reported substantial yields of 4.1 g/L [15] and 22.5 g/L [16] through fermentation in recombinant yeast. However, most studies have focused on the introduction or multiple integration of the tyrosine pathway/phenylalanine pathway, enhancing the precursor’s supply, eliminating phenylalanine or competing tyrosine pathways, and optimizing components of the growth medium or fed-batch fermentation to improve resveratrol in recombination strains [14,15,16]. There are only a few reports on industrial yeast producing resveratrol [10,11].

Rice wine, as an alcoholic beverage, is fermented from glutinous rice using various starter cultures with unique aroma, subtle flavor, and low alcohol content, and is commonly consumed in Asia, including China, Korea, Japan, and India. Rice wine also contains nutritional substances and heath active ingredients, e.g., polyphenols, polysaccharides, peptides, organic acids, trace elements, and other health-promoting compounds [17,18]. However, the content of polyphenols is still lower than fruit wines and juices and tea beverages. *S. cerevisiae* is an important strain for rice wine fermentation, mainly responsible for the production of ethanol, improving the quality of fermentation and preventing spoilage. Additionally, *S. cerevisiae* has a long history of application in food or beverage production, and has been extensively used as a popular host for sustainable production of non-native chemicals and plant-derived compounds with added value in food, pharmaceutical industries, and fuels [19,20,21]. To date, no attempts have been made to produce resveratrol in an industrial rice wine strain (*S. cerevisiae*) during fermentation in rice wine.

In this study, rice wine strain *S. cerevisiae* HJ was engineered to produce resveratrol by introducing *Pc4CL* and *VvSTS*. Next, the production of resveratrol was further increased via the multi-copy integration approach, protein fusion, and removal of feedback inhibition. The combinatorial engineering strategy enabled the production of resveratrol in *S. cerevisiae* HJ08, with the resveratrol titer reaching up to 34.22 ± 3.62 mg/L. Finally, the recombinant strain HJ08 was tested for its ability to produce resveratrol in rice wine fermentation using glutinous rice flour as the material (Figure 2).

## 2. Materials and Methods

### 2.1. Strains, Media, and Culture Conditions

*Escherichia coli* DH5α competent cells (Yeasen Biotechnology Co., Ltd., Shanghai, China) were used for molecular cloning, plasmid propagation, and storage. *E. coli* strains were cultured on Luria–Bertani (LB) medium at 37 °C and supplemented with suitable antibiotics (100 mg/L ampicillin). Yeast strains were cultured in yeast extract peptone dextrose (YPD) media at 30 °C. YPD media was composed of 20 g/L glucose, 20 g/L peptone, and 10 g/L yeast extract. The selected YPD medium used was supplemented with 50 mg/L of Nourseothricin (NTC, Nangjing warbio Biotechnology Co., Ltd., Nanjing, China) or 200 mg/L Hygromycin B (Hyg B, Macklin, Shanghai, China) as needed. For the preparation of solid medium, 2% (*w*/*v*) agar was added into liquid LB and YPD.

### 2.2. Plasmids and Strains Construction

All engineered strains and plasmids are summarized in Table 1. *S. cerevisiae* HJ was isolated from alcohol-active dry yeast for yellow rice wine (Angel Yeast Co., Ltd., Yichang, China) and belongs to a wild type haploid strain, which was used as the host strain for resveratrol biosynthesis. The *RtTAL* (P11544) from *Rhodotorula toruloides* (Synonyms *R. gracilis*), *Pc4CL* (P14912.1) from *Petroselinum crispum*, and *VvSTS* (P28343.2) from *Vitis vinifera* encoding genes were codon-optimized for expression in *S. cerevisiae* and synthesized by Sangon Biotech (Sangon Biotech, Shanghai, China) (Appendix A). Expression cassettes with promoters, terminators, and genes were PCR-amplified using high-fidelity DNA polymerase (Yeasen Biotechnology Co., Ltd., Shanghai, China). All products from PCR or restrictive endonuclease were recovered by gel extraction and electrophoresis using a Gel DNA Extraction Mini Kit (Takara Bio Inc., Dalian, China), and were assembled to plasmid p426 (Bio SCI, Hangzhou, China) using a Seamless Cloning and Assembly Kit (Nangjing warbio Biotechnology Co., Ltd., Nanjing, China). Plasmids were extracted with a Plasmid Miniprep Kit (Takara Bio Inc.). Mutants of *ARO4*^K229L^ and *ARO7*^G141S^ were created by overlapping PCR according to the previous report and sequencing of Sangon Biotech. Plasmids p426-*TAL1*, p426-*4CL*, p426-*STS*, p426-*ARO4*, p426-*ARO7*, and p426-*TAL2* were derived from p426. Recombinant plasmids were confirmed by colony-PCR, digestion, and sequencing (Sangon Biotech, Shanghai, China). The list of primers used for PCR and assembly are given in Table 2.

Strains HJ01, HJ02, HJ03, HJ04, and HJ05 were constructed via homologous recombination in the δ-Site region of the genome using Hyg B markers, respectively, for resistance selection. Strains HJ06, HJ07, and HJ08 were constructed via homologous recombination in the 26S rDNA Site region of the genome using NTC markers. All the linear DNA fragments, including homologous arms, selective markers, and expression cassettes were transformed into yeast using the LiAc/SS carrier DNA/PEG3350 method [22]. The recombination strains were selected on a YPD plate supplemented with Hyg B or NTC and incubated at 30 °C until colonies appeared. The transformants were randomly selected clones and verified by PCR amplification.

### 2.3. Strains Cultivated in Shaking Flasks

The *S. cerevisiae* was inoculated in 50 mL test tubes with 5 mL YPD liquid medium, and cultivated overnight at 30 °C in a rotary shaker (180 rpm). Then, the seed cultures were inoculated in 50 mL of medium with the initial OD_600_ = 0.1–0.2 in 250 mL Erlenmeyer flasks and grown at 30 °C, 180 rpm. Time course experiments were performed to compare the performance of different strains in YPD supplemented with *p*-coumaric acid or tyrosine if needed. Samples were taken at different times for resveratrol production until the resveratrol titer reached a plateau. In addition, the *S. cerevisiae* HJ08 was also tested at a series of tyrosine concentrations: 0 mg/L, 100 mg/L, 200 mg/L, 300 mg/L, and 400 mg/L.

### 2.4. Rice Wine Fermentation with Recombination Strain S. cerevisiae HJ08

A quantity of 100 g glutinous rice flour (obtained from local supermarket) was weighed and mixed with 400 mL water (1:4, *w*/*v*), followed by adding 100 uL α-amylase (Sunson Industry Group Co., Ltd., Beijing, China) for starch liquefaction of glutinous rice flour and incubating at 90 °C for 50 min. Then, 100 uL amyloglucosidase (Sunson Industry Group Co., Ltd.) and 25.0 mg acid protease (Sunson Industry Group Co., Ltd.) were added to the glutinous rice flour solution after cooling to room temperature. The fermentation broth was saccharified at 60 °C for 150 min. Finally, the mixture was sterilized at 121 °C for 15 min, and fermentation was performed by adding 3% (10^6^ CFU/mL) yeast cells (*S. cerevisiae* HJ08) with or without 400 mg/L tyrosine, incubated at 30 °C for 7 days. Samples were collected and centrifuged at 8000× *g* for 10 min to determine the total reducing sugar and resveratrol of the supernatant rice wine.

### 2.5. Analytical Methods

Cell growth was evaluated by OD_600_ using a Shimadzu UV-1900i spectrophotometer (SpectraMax i3, Molecular Devices, Silicon Valley, CA, USA). Total reducing sugar of the supernatant rice wine was measured using DNS based on the method of Khatri et al. [23]. The analysis of resveratrol and *p*-coumaric acid was performed as described by Wang et al. with some modifications [24]. Resveratrol and *p*-coumaric acid were extracted twice and vortexed thoroughly using pure ethyl acetate. After centrifugation (8000× *g*, 10 min, 4 °C), the organic phase was evaporated to dryness, and then the residues were re-dissolved in 1.0 mL pure methanol. Samples were filtered using a 0.22 μm filter membrane, and analyzed by high-performance liquid chromatography (HPLC) analysis, using an Agilent 1260 series instrument (Agilent, Santa Clara, CA, USA) with an Agilent Eclipse Plus C18 column (4.6 × 250 mm, 5 μm). The column was kept at 30 °C and the injection volume was 10 μL. Resveratrol and *p*-coumaric acid were separated by elution with an acetonitrile-acetic acid (1%) (65:35, *v*:*v*) gradient at a flow rate of 0.8 mL/min. Resveratrol was detected at 5.9 min (absorbance at 306 nm) and *p*-coumaric acid was detected 4.2 min (absorbance at 306 nm).

## 3. Results and Discussion

### 3.1. Construction of the Resveratrol Biosynthesis Pathway in S. cerevisiae HJ

Many studies have focused on heterologous resveratrol production via introducing the *4CL* gene from *Arabidopsis thaliana* and *STS* gene *Vitis vinifera* in prokaryotes and eukaryotes [13,25,26]. In this study, the two essential enzyme *4CL* genes from *Petroselinum crispum* and the *STS* gene *Vitis vinifera* were transferred into *S. cerevisiae* HJ01 and integrated into the δ-Site locus of the yeast genome. The resulting strain (HJ01) could produce 0.6 ± 0.01 mg/L resveratrol with 0.5 M *p*-coumaric acid present after 72 h of fermentation. In addition, the best resveratrol production was also determined with different concentrations of *p*-coumaric acid and fermentation time. Similarly, the previous expression of biosynthesis pathway genes from different plant sources using episomal plasmids also resulted in the low titers of resveratrol (from 0.29 mg/L to 3.3 mg/L) by *S. cerevisiae* or *Y. lipolytica* [14,27,28].

### 3.2. Evaluating Resveratrol Biosynthesis Fused Enzymes of Pc4CL-VvSTS

Protein fusion has been widely applied in protein purification, imaging, and drug targeting, and may offer many functions, including improving biological activity of enzymes, facilitating substrate trafficking, and increasing expression yield [29]. It has been reported to be successful in improving the efficiency of substrate delivery to support resveratrol production by employing the fusion protein *4CL*::*STS* with a rigid or flexible linking peptide [24]. Therefore, *VvSTS* was fused to the C-terminus of *Pc4CL* with a rigid (TPTP, EAAAK) linking peptide or a direct fusion in *S. cerevisiae* HJ. *S. cerevisiae* HJ02, *S. cerevisiae* HJ03, and *S. cerevisiae* HJ04 were constructed by integrating at the δ locus sites of *S. cerevisiae* HJ. As shown in Figure 3, the EAAAK linker achieved better results. The HJ03 strain expressing the EAAAK fusion was the highest titer of 34.22 ± 3.62 mg/L, which was increased 57-fold compared with the strain HJ01. In this study, the results showed that the EAAAK linker achieved better results than direct fusion or TPTP linker fusion.

There are two categories of linkers, namely flexible linkers and rigid linkers, which have been applied in linking two enzymes to form unnatural fusion. A suitable linker can improve catalytic efficiency through maintaining the optimal distance between the catalytic centers of enzymes [13,29]. Rigid linker peptides (TPTP, EAAAK) can completely isolate fused enzymes and maintain their independent functions. A previous study reported a similar pattern in which the strain expressing the EAAAK fusion can achieve higher accumulation of resveratrol (1.4-fold) in *Y. lipolytica* than other partial linkers [16]. However, previous studies have also found that fusion using the GSG linker or GGS linker resulted in a 20–50% increase in resveratrol production [13,16,24]. In addition, using synthetic protein scaffolds in yeast and coiled-coil interaction-based enzyme clustering in *E. coli* can increase the production of resveratrol. Synthetic scaffolds increased resveratrol biosynthesis in engineered yeast cells [30].

### 3.3. Alleviation of Tyrosine Feedback Inhibition for Further Enhancement of the Resveratrol Production

Previous studies have shown that the production of amino acids can be increased through the feedback inhibition of key enzymes in *E. coli* or *S. cerevisiae* [31]. According to the experience in previous studies, strategies have been developed to mutate DAHP (3-deoxy-D-arabino- heptulosonate-7-phosphate) synthase *ARO4* and chorismite mutase *ARO7* to eliminate feedback inhibition, which significantly increased the production of aromatic amino acids or derived phenyl propanoic acids in microorganisms. Additionally, it has been also reported that introduction of the mutant alleles *ARO4* and *ARO7* has been shown to increase the production of resveratrol in *S. cerevisiae* and *Y. lipolytica* [14,16].

To test the potential impacts of *ARO4* and *ARO7* alleles on resveratrol synthesis, *S. cerevisiae* HJ06, *S. cerevisiae* HJ07, and *S. cerevisiae* HJ08 were constructed by integrating either a single allele or both alleles, *ARO4*^K229L^ and *ARO7*^G141S^, with the *TAL* pathway at the 26s rDNA region of *S. cerevisiae* HJ03 (Figure 3). The recombinant strain groups of HJ06, HJ07, and HJ08 showed a significantly increase in the average production of resveratrol, by 11-fold (17.08 mg/L), 14-fold (22.09 mg/L), and 22-fold (34.22 mg/L) in comparison with resveratrol production of the parental strain HJ05, respectively. Many researchers have reported that relieving feedback inhibition regulation could enhance the accumulation of *p*-coumaric acid or resveratrol production [32,33].

### 3.4. Batch Fermentation for Resveratrol Production

To determine the optimal precursor supply concentration and fermentation times of the strain HJ08, flask culture fermentation with the recombinant yeast strain was performed in YPD medium supplemented with different concentrations of tyrosine (0, 100, 200, 300, and 400 mg/L) and fermentation different times (24, 48, 72, 96, 120, and 144 h). The strain HJ08 produced 34.22 ± 3.62 mg/L resveratrol at 96 h without tyrosine present, which was a 57-fold increase as compared to that of the strain HJ01 (Figure 4). However, the strain HJ08 only produced 18.62 ± 1.15 mg/L resveratrol with 400 mg/L tyrosine present. The possible reason for this may be the different strain backgrounds and regulation networks of yeast.

The optimization and balancing of microbial growth and product formation have been identified as essential parameters for increasing resveratrol production [10,11,15,16]. The impact of cultivation conditions, namely substrate concentration (glucose), precursor feeding (*p*-coumaric acid, L-Try or L-Phe), and dissolved oxygen concentration, show a huge impact on resveratrol biosynthesis. As illustrated in Figure 5, the OD_600_ of the strain HJ08 reaches a higher value at 72 h of fermentation, leading to a significant increase in resveratrol production in the fermentation solution from 72 h to 96 h. It is well known that the resveratrol stability in solution is affected by high oxygen concentrations and light. This behavior has also been reported with high oxygen concentrations in the bioreactor, which affected negatively the resveratrol titers with *C. glutamicum*, and can lead to the oxidative degradation of excess of resveratrol product [34]. In addition, resveratrol itself could also be toxic to the microbial cell at high concentrations. All these may lead to a reduction in the resveratrol content in the fermentation solution after 120 h.

Resveratrol has been detected in a wide range of plant species, representing 34 families and 100 species. The presence of resveratrol has been demonstrated mainly in the skins or the surface of these plants, and it is also found in a significant number of nutritional foods such as grapes, mulberry, blueberry, peanut, and purple grape juice. It has been estimated that the concentration of resveratrol in fresh grape skin is about 5–10 × 10^−2^ g kg^−1^. Red wine is the major dietary source of resveratrol, in which the concentrations of resveratrol varied from 1.5 to 3 mg L^−1^, while some studies propose higher levels (4–20 mg L^−1^) [3]. In addition, a positive effect of resveratrol has also been observed in humans. According to research, the recommended daily dose should be 12.5 mg/kg body weight, which, according to the levels of resveratrol found in foods, cannot be achieved with wine or any other food [35].

The cell growth of engineered strain HJ08 was also tested in YPD medium supplemented without and with 400 mg/L tyrosine (Figure 5). There were no significant differences in the growth after 144 h of fermentation, which means the addition of tyrosine did not affect cell growth. Furthermore, the addition of 400 mg/L of tyrosine lowered the amount of resveratrol produced after 24 to 144 h of growth in YPD medium.

Demonstration of de novo production of resveratrol in rice wine strain *S. cerevisiae* HJ represents an important step towards commercial production of plant-derived polyphenol with pharma- or nutraceutical properties from glucose. However, commercial production of resveratrol rice wine for direct consumer applications will require substantial further improvement of the product yield and titer by the engineered strains, considering our titer was still lower than the resveratrol titer (4.1 g/L in *S. cerevisiae* and 12 g/L in *Y. lipolytica*) of previous reports [14,16]. Strain improvements are still required to increase resveratrol production yields via combining strategies, including overexpression of pathway genes to improve metabolic flux to aromatic amino acids, increasing the availability of malonyl-CoA, deletion of by-pathway genes for further research [11], and engineering a *S. cerevisiae* coculture platform for the production of flavonoids [36].

### 3.5. Resveratrol Production in Rice Wine Brewing

To directly test the engineered yeast in rice wine production, the recombinant wine yeast strain HJ08 was used for rice wine production in glutinous rice flour liquid broth with or without adding 400 mg/L tyrosine. During 7 days fermentation at 30 °C, the concentration of resveratrol increased slowly throughout the whole brewing process for the HJ08 strain. At the end of fermentation, the resveratrol content of the rice wine fermented with engineered strain HJ08 reached 2.04 ± 0.08 mg/L and 1.45 ± 0.06 mg/L with or without adding 400 mg/L tyrosine, respectively (See Appendix A). The result suggested that resveratrol could be produced in rice wine using the strain HJ08. A previous study reported that the content of resveratrol was 3.44 mg/L and 0.68 mg/L in grape juice or white wine production using engineered yeast [37]. When using *p*-coumaric acid as a precursor, the resveratrol content of the red wine was 0.4028 mg/L using the engineered strain EC118 after 7 days of fermentation at 28 °C [38]. Another study using engineered *S. cerevisiae* YS58 as the host strain for resveratrol production found the resveratrol content in grape wine reached 31 ± 2 μg/L after 19 days of fermentation [39]. All these results indicated that controlled fermentation using engineered yeast strains for enhancing functional components has attracted attention in the wine industry.

## 4. Conclusions

The recently increased high demand for resveratrol inspired research on the development of microbial cell factories for its heterologous production. In this study, we explored the possibility of resveratrol production from glucose as the sole carbon source through the genome integration of the resveratrol biosynthetic pathway using the rice wine yeast *S. cerevisiae* HJ, and the optimization of the strain to reach a resveratrol titer of 34.22 ± 3.62 mg/L. In addition, the recombination strain HJ08 was able to increase the content of resveratrol in rice wine, and the titer of resveratrol reached 2.04 ± 0.08 mg/L. We hope this approach can lay a foundation for resveratrol application in the food industry for bioactive compounds and broaden the scope of other chemical production in yeast.

## Figures and Tables

**Figure 1 jof-10-00513-f001:**
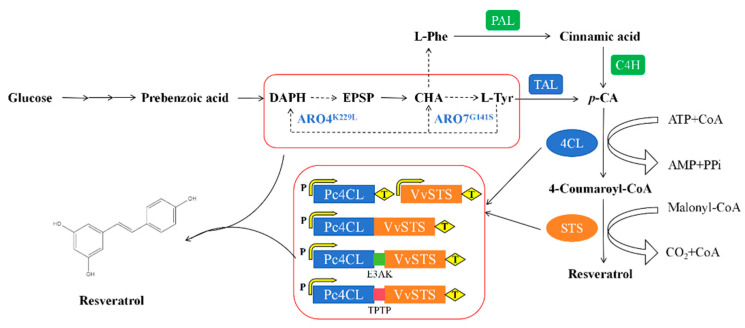
Genes and enzymes involved in the biosynthetic pathways of resveratrol in engineered rice wine strain HJ. *DAHP*, 3-deoxy-arabino-heptulonate-7-phosphate; *EPSP*, 5-enolpyruvyi -shikimate 3-phosphate; *CHA*, chorismic acid; L-Phe, L-phenylalanine; L-Tyr, L-tyrosine; *p*-CA, *p*-Coumaric acid; *ARO4*, 3-deoxy-7-phosphoheptulonate synthase; *ARO7*, Chorismite mutase; *TAL*, tyrosineammonia lyase; *4CL*, 4-coumarate-CoA ligase; *STS*, resveratrol synthase; *PAL*, phenylalanine ammonia lyase; *C4H*, *trans*-cinnamate 4-monooxygenase.

**Figure 2 jof-10-00513-f002:**
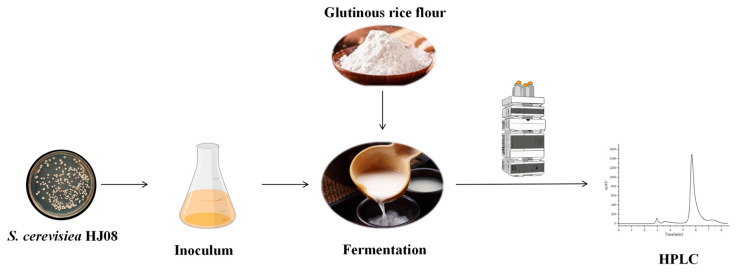
Schematic diagram of resveratrol rice wine fermentation by engineered strain HJ08.

**Figure 3 jof-10-00513-f003:**
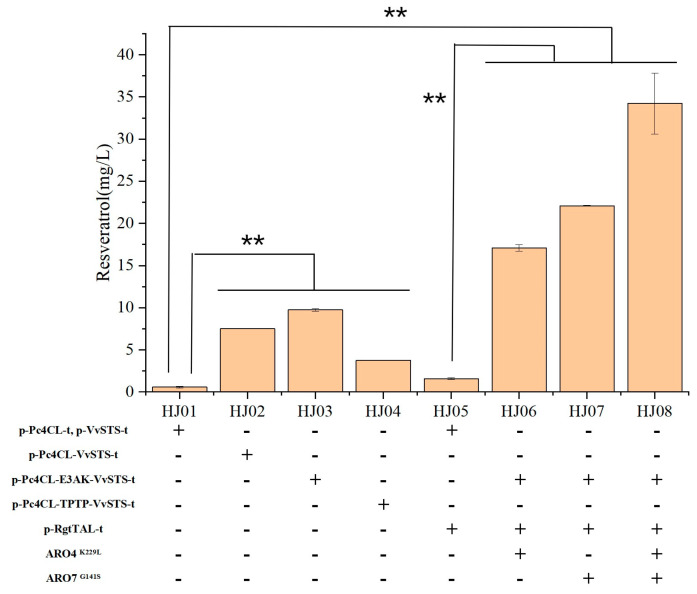
Effect of gene fusion and the elimination of the feedback inhibition of *ARO4*^K229L^ and *ARO7*^G141S^ on resveratrol production. Values and error bars represent the standard errors of the average reading and standard deviations for three replicates. ‘-’ symbol means without genetic modification, and ‘+’ means with genetic modification. The resveratrol production was measured after the strains were cultured in shake flasks using YPD medium for 96 h. Statistical analysis was performed using Student’s *t*-test (two-tailed; two-sample assuming equal variance; ** *p* < 0.01) focused on the resveratrol production. The displayed average values and standard deviations were calculated from three independent biological experiments.

**Figure 4 jof-10-00513-f004:**
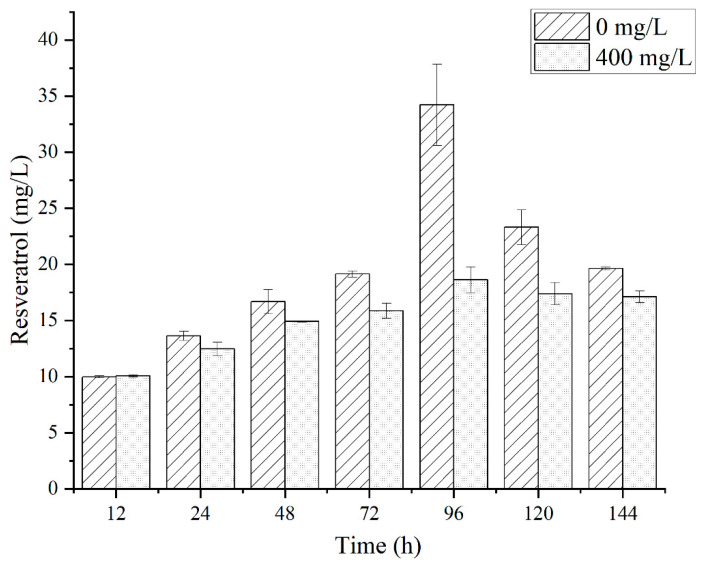
Resveratrol production of engineered strain HJ08 in YPD medium with (400 mg/L) and without tyrosine at different times. The resveratrol production was measured after the strains were cultured in shake flasks using YPD medium for 96 h. Values and error bars represent the standard errors of the average reading and standard deviations for three replicates.

**Figure 5 jof-10-00513-f005:**
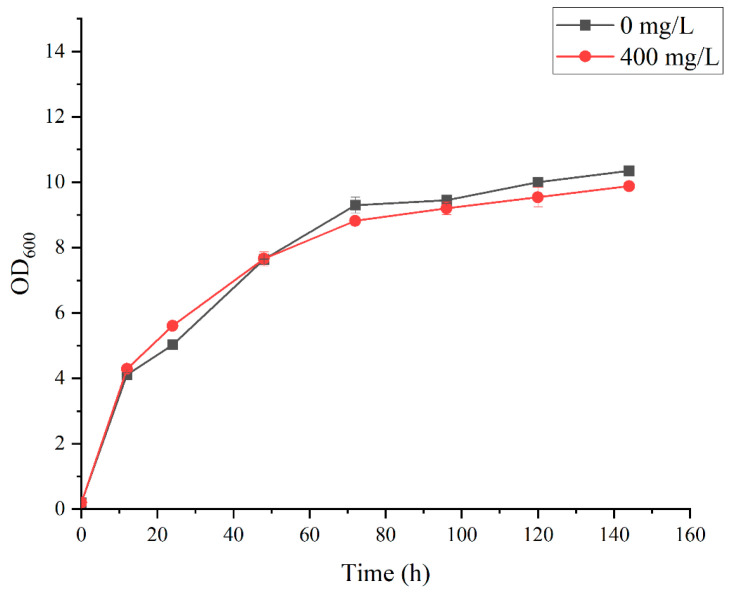
Growth profile of resveratrol-producing strain HJ08 in YPD medium with 400 mg/L (●) and without tyrosine (■). Values and error bars represent the standard errors of the average reading and standard deviations for three replicates.

**Table 1 jof-10-00513-t001:** Strains and plasmids used in this study.

Strains and Plasmids	Relevant Characteristics	Source
*S. cerevisiae* strains		
HJ	wild type haploid strain	Angel, China
HJ01	HJ, δ-Site:: P*_GPD1_-Pc4CL-*T*_ADH1_*, P*_TEF1_-VvSTS-*T*_ADH2_*, Hygromycin B marker	This study
HJ02	HJ, δ-Site:: P*_GPD1_-Pc4CL*-*VvSTS-*T*_ADH2_*, Hygromycin B marker	This study
HJ03	HJ, δ-Site:: P*_GPD1_-Pc4CL*-E3AK-*VvSTS-*T*_ADH2_*, Hygromycin B marker	This study
HJ04	HJ, δ-Site:: P*_GPD1_-Pc4CL*-TPTP-*VvSTS-*T*_ADH2_*, Hygromycin B marker	This study
HJ05	HJ, δ-Site:: P*_PGK1_-*Rt*TAL-*T*_CYC1_*, P*_GPD1_-Pc4CL-*T*_ADH1_*, P*_TEF1_-VvSTS-*T*_ADH2_*, Hygromycin B marker	This study
HJ06	HJ04, rDNA site:: P*_GPM1_*-Rt*TAL*-T*_FBA1_*, P*_PGK1_*-*ARO4*^K229L^-T*_CYC1_*, clonNAT marker	This study
HJ07	HJ04, rDNA site:: P*_GPM1_*-Rt*TAL*-T*_FBA1_*, P*_TEF1_*-*ARO7*^G141S^-T*_ADH1_*, clonNAT marker	This study
HJ08	HJ04, rDNA site:: P*_GPM1_*-Rt*TAL*-T*_FBA1_*, P*_PGK1_*-*ARO4*^K229L^-T*_CYC1_*, P*_TEF1_*-*ARO7*^G141S^-T*_ADH1_*, clonNAT marker*_,_*	This study
Plasmids		
pAG36	CEN/ARS, P*_AgTEF1_*-natMX-T*_AgTEF1_*, Amp^r^, clonNAT resistance	Euroscarf, Frankfurt, Germany
pUG27	loxP-P*_AgTEF1_*-Sphis5-T*_AgTEF1_*-loxP, Amp^r^, histidine prototrophy	Euroscarf, Frankfurt, Germany
pUG75	loxP-P*_AgTEF1_*-hphMX-T*_AgTEF1_*-loxP, Amp^r^, Hygromycin B resistance	Euroscarf, Frankfurt, Germany
p426	2 μm ori, *URA3*, P*_PGK1_*-T*_CYC1_*, Amp^r^	Bio SCI, Hangzhou, China
pUG27-NTC	clonNAT resistance, Amp^r^, clonNAT resistance as a selection marker by δ-integration	This study
p426-*TAL1*	2 μm ori, *URA3*, P*_PGK1_-*RtTAL*-*T*_CYC1_*	This study (Appendix A)
p426-*4CL*	2 μm ori, *URA3*, P*_GPD1_-Pc4CL-*T*_ADH1_*	This study (Appendix A)
p426-*STS*	2 μm ori, *URA3*, P*_TEF1_-VvSTS-*T*_ADH2_*	This study (Appendix A)
p426-*ARO4*	2 μm ori, *URA3*, P*_PGK1_*-*ARO4*^K229L^-T*_CYC1_*	This study
p426-*ARO7*	2 μm ori, *URA3*, P*_TEF1_*-*ARO7*^G141S^-T*_ADH1_*	This study
p426-*TAL2*	2 μm ori, *URA3*, P*_GPM1_*-Rt*TAL*-T*_FBA1_*	This study (Appendix A)

**Table 2 jof-10-00513-t002:** Primers used in this study.

Primers	Sequences (5′-3′)	Applications
**Amplification and δ-Site integration of *4CL* and *STS* genes in HJ01**		
1	CTCGAGGGATATAGGAATCCTC	Forward primer for amplification of δ-Site upstream region
2	TATTGATAATGATAAACTCGAACTGTGTTGGAATAGAAATCAACTATCATC	Reverse primer for amplification of δ-Site upstream region
3	GATGATAGTTGATTTCTATTCCAACACAGTTCGAGTTTATCATTATCAATA	Forward primer for amplification of *4CL* from p426-*4CL*
4	AAACATTTTGAAGCTATGGTGTGTGGATCCGTGTGGAAGAACGATTACAA	Reverse primer for amplification of *4CL* from p426-*4CL*
5	TTGTAATCGTTCTTCCACACGGATCCACACACCATAGCTTCAAAATGTTT	Forward primer for amplification of *STS* from p426-*STS*
6	TAAGGGTTGTCGACCTGCAGCGTAGGCCGCAAATTAAAGCCTTC	Reverse primer for amplification of *STS* from p426-*STS*
7	TATTTCTCATTTTCCTTCGCATGCCTACGCTGCAGGTCGACAACCCTTAA	Forward primer for amplification of HygB selected maker from pUG75
8	AACAACACCTGCTTCATCAGCTGTTACGACTCACTATAGGGAGACCGGCA	Reverse primer for amplification of HygB selected maker from pUG75
9	TGCCGGTCTCCCTATAGTGAGTCGTAACAGCTGATGAAGCAGGTGTTGTT	Forward primer for amplification of δ-Site downstream region
10	GAGAACTTCTAGTATATTCTGTATACCTAATATT	Reverse primer for amplification of δ-Site downstream region
**Amplification and δ-Site integration of *4CL* and *STS* genes in HJ02 (Directed fusion) using primers 1/2, 3/11, 12/6, 7/8, 9 and 10**		
11	TTCTGAATTCTTCAACGGAAGCCATCTTTGGCAAGTCACCAGAAGCGATC	Reverse primer for amplification of *4CL* from p426-*4CL*
12	GATCGCTTCTGGTGACTTGCCAAAGATGGCTTCCGTTGAAGAATTCAGAA	Forward primer for amplification of *STS* from p426-*STS*
**Amplification and δ-Site integration of *4CL* and *STS* genes in HJ03 (E3AK linker) using primers 1/2, 3/13, 14/6, 7/8, 9 and 10**		
13	TCTTCAACGGAAGCCATCTTAGCAGCAGCTTCCTTTGGCAAATCACCAGA	Reverse primer for amplification of *4CL* from p426-*4CL*
14	TCTGGTGATTTGCCAAAGGAAGCTGCTGCTAAGATGGCTTCCGTTGAAGA	Forward primer for amplification of *STS* from p426-*STS*
**Amplification and δ-Site integration of *4CL* and *STS* genes in HJ04 (TPTP linker) using primers 1/2, 3/15, 16/6, 7/8, 9 and 10**		
15	GAATTCTTCAACGGAAGCCATAGGTGTAGGTGTCTTTGGCAAATCACCAGA	Reverse primer for amplification of *4CL* from p426-*4CL*
16	TCTGGTGATTTGCCAAAGACACCTACACCTATGGCTTCCGTTGAAGAATTC	Forward primer for amplification of *STS* from p426-*STS*
**Amplification and δ-Site integration of *TAL*, *4CL* and *STS* genes in HJ05 using primers 1/17, 18/19, 20/4, 5/6, 7/8, 9 and 10**		
17	CTACGTAAGATAATTGTATATTACGCAGCTGAAGCTTCGTACGC	Reverse primer for amplification of δ-Site upstream region
18	GATGATAGTTGATTTCTATTCCAACATATTTTAGATTCCTGACTTCAACTC	Forward primer for amplification of *TAL* from p426-*TAL1*
19	TATCAGATCCACTAGTGGCCTATGCAGTGTCGAAAACGAGCTCAGT	Reverse primer for amplification of *TAL* from p426-*TAL1*
20	GGGACGCTCGAAGGCTTTAATTTGCCAGTTCGAGTTTATCATTATCAATA	Forward primer for amplification of *4CL* from p426-*4CL*
**For site-directed mutagenesis of *ARO4* and *ARO7* genes**		
21	ACAAATATAAAAACAACTAGTGGATCCATGAGTGAATCTCCAATGTTCG	Amplify *ARO4* for p426-*ARO4* construction from *S. cerevisiae* genome, forward primer.
22	TCGAGGTCGACGGTATCGATAAGCTTCATTTCTTGTTAACTTCTCTTC	Amplify *ARO4* for p426-*ARO4* construction from *S. cerevisiae* genome, reverse primer.
23	CATTTCATGGGTGTTACTTTGCATGGTGTTGCTGCTATC	*ARO4* site-directed mutations primers, forward primer.
24	GATAGCAGCAACACCATGCAAAGTAACACCCATGAAATG	*ARO4* site-directed mutations primers, reverse primer.
25	CTAAGTTTTAATTACAAAACTAGTATGGATTTCACAAAACCAGAAAC	Amplify *ARO7* for p426-*ARO7* construction from *S. cerevisiae* genome, forward primer.
26	GAATTCCTGCAGCCCGGGGGATCCTCACTCTTCCAACCTTCTTAGCAAG	Amplify *ARO7* for p426-*ARO7* construction from *S. cerevisiae* genome, reverse primer.
27	GATAAGAATAACTTCAGTTCTGTTGCCACTAG	*ARO7* site-directed mutations primers, forward primer.
28	CTAGTGGCAACAGAACTGAAGTTATTCTTATC	*ARO7* site-directed mutations primers, reverse primer.
**Amplification and rDNA-Site integration of *TAL* and *ARO4*** ^K229L^ ** genes in HJ06 using primers 30/31, 32/33, 34/35, 36/37, 38 and 39**		
30	CCGGAACCTCTAATCATTCG	Forward primer for amplification of rDNA-Site upstream region
31	CTTAAAGTCATACATTGCACGACTAGCTTGAGGTATAATGCAAGTACGGT	Reverse primer for amplification of rDNA-Site upstream region
32	ACCGTACTTGCATTATACCTCAAGCTAGTCGTGCAATGTATGACTTTAAG	Forward primer for amplification of *TAL* from p426-*TAL2*
33	GAGTTGAAGTCAGGAATCTAAAATAAAAGATGAGCTAGGCTTTTGTAAAA	Reverse primer for amplification of *TAL* from p426-*TAL2*
34	TTTTACAAAAGCCTAGCTCATCTTTTATTTTAGATTCCTGACTTCAACTC	Forward primer for amplification of *ARO4* ^K229L^ from p426-*ARO4*
35	TTAAGGGTTGTCGACCTGCAGCGTAGCAAATTAAAGCCTTCGAGCGTCCC	Reverse primer for amplification of *ARO4* ^K229L^ from p426-*ARO4*
36	GGGACGCTCGAAGGCTTTAATTTGCTACGCTGCAGGTCGACAACCCTTAA	Forward primer for amplification of NTC selected maker from pUG27-NTC
37	TCTCTGCGTGCTTGAGGTATAATGCACGACTCACTATAGGGAGACCGGCA	Reverse primer for amplification of NTC selected maker from pUG27-NTC
38	TGCCGGTCTCCCTATAGTGAGTCGTGCATTATACCTCAAGCACGCAGAGA	Forward primer for amplification of rDNA-Site downstream region
39	AACGAACGAGACCTTAACCT	Reverse primer for amplification of rDNA-Site downstream region
**Amplification and rDNA-Site integration of *TAL* and *ARO7*^G141S^ genes in HJ07 using primers 30/31, 32/40, 41/42, 43/37, 38 and 39**		
40	AAACATTTTGAAGCTATGGTGTGTGAAAGATGAGCTAGGCTTTTGTAAAA	Reverse primer for amplification of *TAL* from p426-*TAL2*
41	TTTTACAAAAGCCTAGCTCATCTTTCACACACCATAGCTTCAAAATGTTT	Forward primer for amplification of *ARO7* ^G141S^ from p426-*ARO7*
42	TTAAGGGTTGTCGACCTGCAGCGTAGATCCGTGTGGAAGAACGATTACAA	Reverse primer for amplification of *ARO7* ^G141S^ from p426-*ARO7*
43	TTGTAATCGTTCTTCCACACGGATCTACGCTGCAGGTCGACAACCCTTAA	Forward primer for amplification of NTC selected maker from pUG27-NTC
**Amplification and rDNA-Site integration of *TAL*, *ARO4*** ^K229L^ ** and *ARO7*** ^G141S^ ** genes in HJ08 using primers 30/31, 32/33, 34/44, 45/42, 43/37, 38 and 39**		
44	AAACATTTTGAAGCTATGGTGTGTGGCAAATTAAAGCCTTCGAGCGTCCC	Reverse primer for amplification of *ARO4* ^K229L^ from p426-*ARO4*
45	GGGACGCTCGAAGGCTTTAATTTGCCACACACCATAGCTTCAAAATGTTT	Forward primer for amplification of *ARO7* ^G141S^ from p426-*ARO7*

## Data Availability

The original contributions presented in the study are included in the article/Appendix A, further inquiries can be directed to the corresponding authors.

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
