# Peer review of "Denovo Production of Resveratrol by Engineered Rice Wine Strain Saccharomyces cerevisiae HJ08 and Its Application in Rice Wine Brewing"

_jof, 2024, doi:10.3390/jof10080513_

Round 1

Reviewer 1 Report

Manuscript jof-3086807 " Denovo production of resveratrol by engineered Rice Wine Strain Saccharomyces cerevisiae HJ08 and Its Application in Rice Wine Brewing" by An, H., et.al. describes installation of a resveratrol biosynthetic pathway into an industrial S. cerevisiae strain used in rice wine fermentation processes. The goal of this project would be generating a process to produce rice wine that has the benefit of resveratrol supplementation. This compound has a variety of claimed health benefits and naturally supplements grape and some other fruit wines owing to the presence of resveratrol in grape skins and some other fruit sources. The authors constructed codon optimized synthetic genes that could be integrated at rDNA sites and delta elements in the S. cerevisiae genome. Hygromycin and Nourseothricin resistance genes were used for selection markers as the recipient strain lacked any auxotrophic markers. The investigators demonstrated resveratrol production by the engineered strain and found maximal production was achieved in a strain expressing a Pc 4CL - Vv STS fusion linked by a EA3K linker. Feedback insensitive Aro4 and Aro7 further improved resveratrol synthesis. The authors show that tyrosine addition did not improve resveratrol production during fermentation in YPD medium. Finally, the authors report resveratrol production during fermentation of saccharified rice flour.

Comments

Figure 3 should include an indication of whether there are statistically significant differences among the resveratrol yields. There should also be an indication in the figure legend and/or the methods section describing what statistical test was used to determine significance.

The time frame for the fermentation results shown in Figure 3 are not clearly indicated. Is this a 96 hour culture in YPD?

In Figure 3 the strain HJ08 achieved 34.22 ±3.62 mg/L resveratrol. Is this the same data used in Figure 4 at the 96 hour time point which is reported as 34.22±3.62 mg/L?

The very large increase in resveratrol from 72 - 96 hours followed by significant reduction at 120 hours is worth some explanation or speculation. Is the product being metabolized by the cells? Would we expect a similar effect in a rice wine fermentation? Some consideration of this point in the discussion would be helpful since this will be a very important point for any industrial process using the strain.

Section 3.4 describes fermentation reactions in YPD with various concentrations of tyrosine but only ± 400 mg/L is shown. This should be clarified in the text minimally by the addition of (data not shown) but it would be more useful to show the data?

A Rice flour fermentation is described in methods and section 3.5 described fermentation in saccharified rice flour medium and the result is reported in the abstract and text but there are no data figures presented that uses rice flour fermentation. It would be useful to present these data as more than the HPLC traces shown in the supplemental file.

Overall, the authors show no improvement in resveratrol production over production by other S. cerevisiae strains. There are no novel engineering or fermentation strategies applied as the fusion protein strategy has been applied before in different yeast strains with the same results. This issue is nonetheless of interest from the perspective of being able to supplement rice wine with a potentially useful compound.

Abstract line 24 "... with or without addition of 400 mg/L after 7 days."  Addition of what? should this say "...400mg/L tyrosine..."

Figure 1 is not referenced anywhere in the text. The figure itself is not clear as it has resveratrol production on both sides of the figure.

The reference to figure numbers in the text are mixed up in several places, for example Line 189 - figure 2 is actually describing figure 3.

Author Response

Comments 1: Figure 3 should include an indication of whether there are statistically significant differences among the resveratrol yields. There should also be an indication in the figure legend and/or the methods section describing what statistical test was used to determine significance.

Response 1: Thank you for pointing this out, and we agree this is an excellent suggestion. Therefore, we reprocessed the data and conducted statistical analysis using Student's t-test. Detailed experimental results and explanations have been indicated in Figure 3 (Line 226 and red section in line 231).

Comments 2: The time frame for the fermentation results shown in Figure 3 are not clearly indicated. Is this a 96 hour culture in YPD?

Response 2: Thank you for your reminder, and we were really sorry for our careless. As you mentioned, the result was achieved through fermentation in YPD medium for a duration of 96 h. This information has been incorporated into the description provided in Figure 3 (Line 230).

Comments 3: In Figure 3 the strain HJ08 achieved 34.22 ±3.62 mg/L resveratrol. Is this the same data used in Figure 4 at the 96 hour time point which is reported as 34.22±3.62 mg/L?

Response 3: Thank you for your inquiry, we apologize for any confusion caused by our lack of clarity. We have provided annotations for Figure 3 (Line 230) and Figure 4 (Line 256) to enhance your understanding of them, which indicated the resveratrol production are both 34.22±3.62 mg/L.

Comments 4: The very large increase in resveratrol from 72 - 96 hours followed by significant reduction at 120 hours is worth some explanation or speculation. Is the product being metabolized by the cells? Would we expect a similar effect in a rice wine fermentation? Some consideration of this point in the discussion would be helpful since this will be a very important point for any industrial process using the strain.

Response 4: We feel great thanks for your professional review work on our article. According to your nice suggestion, we have incorporated additional content into the article, including the exact location where the change can be found in the revised manuscript in line 243. The concentrations of resveratrol produced by this engineered yeast strain during rice wine production are similar to the range present in others reports. Actually, the engineered strain are still required to increase resveratrol production yields via combining strategies. In addition, the optimization and balancing of microbial growth and product formation have been identified as essential parameters for increasing resveratrol production, it is need to further investigate the precursor concentration, inoculum volume, and even co-culture with multiple strains capable of metabolizaing different substrates or producing different prodcuts.

Comments 5: Section 3.4 describes fermentation reactions in YPD with various concentrations of tyrosine but only ± 400 mg/L is shown. This should be clarified in the text minimally by the addition of (data not shown) but it would be more useful to show the data?

Response 5: Thank you very much for your advice. We have carefully selected the optimal amount of tyrosine to be added in order to present the final experimental results. As a result, we have included "data not shown" in the article (Line 238). Actually, the engineered strains were grown in 50 mL test tube (10 mL medium) with various concetrations of tyrosine, and selected the concentration and addition time of precursor (L-Tyr).

Comments 6: A Rice flour fermentation is described in methods and section 3.5 described fermentation in saccharified rice flour medium and the result is reported in the abstract and text but there are no data figures presented that uses rice flour fermentation. It would be useful to present these data as more than the HPLC traces shown in the supplemental file.

Response 6: Thank you very much for your advice. In this part, we expect a similar effect in a rice wine fermentation as the result in flask fermentation. However, the concentrations of resveratrol produced by this engineered yeast strain during rice wine production are similar to the range present in others reports. In order to improve the production of resveratrol, the engineered yeast strain are still required to modify via combining strategies. There are much work need to further investigate the fermentation processing.

Comments 7: Overall, the authors show no improvement in resveratrol production over production by other S. cerevisiae strains. There are no novel engineering or fermentation strategies applied as the fusion protein strategy has been applied before in different yeast strains with the same results. This issue is nonetheless of interest from the perspective of being able to supplement rice wine with a potentially useful compound.

Response 7: We feel great thanks for your professional review work on our article. In this study, we explored the possibility of resveratrol production from glucose as sole carbon source through genome integration of resveratrol biosynthetic pathway using the rice wine yeast S. cerevisiae HJ, and increase the content of resveratrol in rice wine.

Comments 8: Abstract line 24 "... with or without addition of 400 mg/L after 7 days."  Addition of what? should this say "...400mg/L tyrosine..."

Response 8: Thanks for your careful checks. We are sorry for our carelessness. Based on your comments, we have made corrections to enhance the comprehensiveness of the article's content (Line 24).

Comments 9: Figure 1 is not referenced anywhere in the text. The figure itself is not clear as it has resveratrol production on both sides of the figure.

Response 9: We sincerely appreciate the valuable comments. We have thoroughly reviewed our article and are confident that we have accurately referenced "Figure 1" within the article (Line 47). Regarding Figure 1, we trust that our explanation will justify the content: the left side represents resveratrol produced by the strain HJ08 modified in our experiment, while the right side depicts resveratrol obtained after transferring key genes in the synthesis pathway.

Comments 10: The reference to figure numbers in the text are mixed up in several places, for example Line 189 - figure 2 is actually describing figure 3.

Response 10: We sincerely thank you for careful reading and feel sorry for our carelessness. In our resubmitted manuscript, this kind of mistake has been revised (Line 189).

In addition, we tried our best to improve the manuscript and made some changes marked in red in revised paper which will not influence the content and framework of the paper. We hope the correction will meet with approval. Once again, thank you very much for your comments and suggestion.

Reviewer 2 Report

In this manuscript, An and co-workers present a study regarding the possibility to use engineered strains of rice wine Saccharomyces cerevisiae for de novo production of resveratrol. For this purpose, the authors integrated 4-coumaroyl-CoA ligase (Pc4CL) from Petroselinum crispum and stilbene synthase (VvSts) from Vitis vinifera in the genome of the industrial rice wine strain of Saccharomyces cerevisiae HJ. The two heterologous genes were integrated in different combinations (individual, fused, fused with spacers in between) to obtain 8 different strains, HJ01-HJ08. The expected resveratrol production during fermentation was improved by removing the feedback inhibition caused by tyrosine through point mutations in ARO4 and ARO7.

The study is timely and of biotechnological relevance, as it aims to optimize the production of resveratrol by engineered microorganisms.

There are some issues that must be addressed before the manuscript can be considered for publication.

-        One major concern is the absence of any data regarding the expression of Pc4CL and VvSts in the S. cerevisiae obtained. The authors are encouraged to determine the expression of Pc4CL and VvSt (individual and fused) at transcript or protein level (or both).

-        Figure 2 is superfluous and does not add much to the text.

-        In Figure 3, a negative control is missing (e.g., HJ, or HJ transformed with empty vector p426). Also, please indicate the time point for resveratrol determination.

-        Figure 4 indicates that resveratrol production is maximum after 96 h. Was this trend noticed for the other strains as well?

-        The authors showed that supplemental tyrosine did not improve resveratrol production by HJ08 strain. Was the situation similar in the case of the other (less producing) strains?

-        Figure 2 is superfluous and does not add much to the text.

Author Response

Comments 1: One major concern is the absence of any data regarding the expression of Pc4CL and VvSts in the S. cerevisiae obtained. The authors are encouraged to determine the expression of Pc4CL and VvSt (individual and fused) at transcript or protein level (or both).

Response 1: We feel great thanks for your professional review work on our article. The concentrations of resveratrol produced by this engineered yeast strain during rice wine production are similar to the range present in others reports. Actually, the engineered strain are still required to increase resveratrol production yields via combining strategies. We will determine the gene expression at transcript or protein level in future study.

Comments 2: Figure 2 is superfluous and does not add much to the text.

Response 2: Thank you for pointing this out. We have provided annotations for Figure 2 (Line 89).

Comments 3: In Figure 3, a negative control is missing (e.g., HJ, or HJ transformed with empty vector p426). Also, please indicate the time point for resveratrol determination.

 Response 3: Thank you for this valuable comment. In this study, we explored the possibility of resveratrol production from glucose as sole carbon source through genome integration of resveratrol biosynthetic pathway using the rice wine yeast S. cerevisiae HJ. There is no negative control(can not produce resveratrol). However, we will perform related studies for deeply and thoroughly understand this problem in the future study. Besides, we feel sorry for our carelessness about the lack of time point for resveratrol determination. The result was achieved through fermentation in YPD medium for a duration of 96 h and this information has been incorporated into the description provided in Figure 3 (Line 230).

Comments 4: Figure 4 indicates that resveratrol production is maximum after 96 h. Was this trend noticed for the other strains as well?

Response 4: Thank you for your inquiry, we feel a bit of pity for this confusion because we did not actually conduct that portion of the study. In the previous study, we identified that the strain HJ08 produced the highest yield of resveratrol by controlling relevant variables. Based on this finding, we investigated the production of resveratrol at different time points to determine the optimal timing and applied it to the fermentation process of rice wine, which is our main focus in fermentation optimization. However, for future studies, we plan to explore using different strains for fermentation optimization at various time points.

Comments 5: The authors showed that supplemental tyrosine did not improve resveratrol production by HJ08 strain. Was the situation similar in the case of the other (less producing) strains?

Response 5: Thank you for this valuable comment. As previous response 4, we only identified that the strain HJ08 produced the highest yield of resveratrol by controlling relevant variables, and plan to explore using different strains for fermentation optimization at various time points for future research.

Comments 6: Figure 2 is superfluous and does not add much to the text.

Response 6: Thank you for pointing this out. We have provided annotations for Figure 2 (Line 89), and think it can explain the rice wine fermentation processing.

We tried our best to improve the manuscript and made some changes marked in red in revised paper which will not influence the content and framework of the paper. We appreciate for your warm work earnestly, and hope the correction will meet with approval. Once again, thank you very much for your comments and suggestion.

Reviewer 3 Report

The presented study is well thought out and well placed in the current context of more and more consumers looking to consume foods that are health promoting.  A bit of caution should be inserted as resveratrol can also become toxic at very high amounts consumed.

Line 71

Replace alcoholicity with alcohol content

Line 72

Rice wine also known for the abundant of nutritional substances and heath active ingredients,

Change to:  Rice wine also contains nutritional substances and heath active ingredients,…

Line 170-172  Cite references here:

Many studies have focused on heterologous resveratrol production via introducing the 4CL gene from Arabidopsis thaliana and STS gene Vitis vinifera in prokaryotes and eukaryotes. 

Line 173

STS gene Vitis vinifera were transformed into S. cerevisiae HJ01 and integrated into the

Change to: STS gene Vitis vinifera were transferred into S. cerevisiae HJ01 and integrated into the

Fig 2

Replace ‘seed solution’ with ‘inoculum’

Line 247

And the addition of 400 mg/L of tyrosine lowered the amount of resveratrol produced after 24 to 144 hours of growth in YPD medium.

Lines 252-258

When discussing the amounts of resveratrol produced by the yeast in their study, the authors should consider what concentrations of resveratrol are found in grapes and other plant foods.  At very high concentrations resveratrol can damage the liver, though the concentrations achieved here appear to be well below any harmful concentration.

The concentrations of resveratrol produced by this engineered yeast strain during rice wine production are similar to the range present in various plant products.  Thus, the amounts of resveratrol produced in this study can be considered an enrichment of the bioactive substances and the nutritional quality of the rice wine produced.

Author Response

Major comments: The presented study is well thought out and well placed in the current context of more and more consumers looking to consume foods that are health promoting.  A bit of caution should be inserted as resveratrol can also become toxic at very high amounts consumed.

Response: We think this is an excellent suggestion.  According to research, the recommended daily dose should be 12.5 mg/kg body weight, which, according to the levels of resveratrol found in food, cannot be achieved with wine or any other food(Weiskirchen, 2016). However, many of the results come from tests on rodents and under modified laboratory conditions.  In order for these effects to be applied in therapy or treatment, the wine would have to be consumed in high amounts. This eliminates health benefits due to the alcohol content. Many scientists are currently working to develop a suitable derivative that would increase bioavailability so that concentrated resveratrol can be used for therapeutic purposes. Even at low doses, resveratrol has been shown to be an excellent cardioprotectant, and has an antidiabetic effect, neuroprotective activity,  and also retards aging.

Comments 1: Line 71 Replace alcoholicity with alcohol content.

Response 1: We sincerely thank you for careful reading. As you suggested, we have replaced alcoholicity with alcohol content (Line 71).

Comments 2Line 72  Rice wine also known for the abundant of nutritional substances and heath active ingredients, Change to:  Rice wine also contains nutritional substances and heath active ingredients,…

Response 2: We think this is an excellent suggestion. We have re-written this part according to your comment, including the exact location where the change can be found in the revised manuscript (Line 72).

Comments 3Line 170-172 Cite references here: Many studies have focused on heterologous resveratrol production via introducing the 4CL gene from Arabidopsis thaliana and STS gene Vitis vinifera in prokaryotes and eukaryotes.

Response 3: We sincerely appreciate the valuable comments. As you suggested, we have cited  references to support this idea (13,25,26).

Comments 4Line 173  STS gene Vitis vinifera were transformed into S. cerevisiae HJ01 and integrated into the  Change to: STS gene Vitis vinifera were transferred into S. cerevisiae HJ01 and integrated into the

Response 4: We sincerely thank you for careful reading. According to your nice suggestion, we have replaced “transformed” with “transferred” (Line 173).

Comments 5Fig 2. Replace ‘seed solution’ with ‘inoculum’

Response 5: Thanks for your careful checks. Based on your comments, we have replaced seed solution with inoculum (Figure 2 in line 192).

Comments 6Line 247 And the addition of 400 mg/L of tyrosine lowered the amount of resveratrol produced after 24 to 144 hours of growth in YPD medium.

Response 6: Based on your comments, we have added this sentence.   

Comments 7Lines 252-258 When discussing the amounts of resveratrol produced by the yeast in their study, the authors should consider what concentrations of resveratrol are found in grapes and other plant foods.  At very high concentrations resveratrol can damage the liver, though the concentrations achieved here appear to be well below any harmful concentration.

The concentrations of resveratrol produced by this engineered yeast strain during rice wine production are similar to the range present in various plant products.  Thus, the amounts of resveratrol produced in this study can be considered an enrichment of the bioactive substances and the nutritional quality of the rice wine produced.

Response 7: We sincerely appreciate the valuable comments. As you suggested, We have re-written this part according to your comment (Line 257).

In addition, we tried our best to improve the manuscript and made some changes marked in red in revised paper which will not influence the content and framework of the paper. We hope the correction will meet with approval. Once again, thank you very much for your comments and suggestion.

Round 2

Reviewer 1 Report

My concerns with the original submission have been addressed. 

Table 2. Primers 3, 5, 7 ,12, 14, 16, 18, 20, 32, 34, 36 ,45 are described in the table as "Ferverse primer..." I think these should be described in the table as "Forward primer..."

Primer 10 in Table 2 is described as "Dorward primer..." I think this should be "Reverse primer..".

Authors should correct these errors

Author Response

Major comments: My concerns with the original submission have been addressed.

Response : Thank you very much for your decision and constructive comments on my manuscript, you have provided us with very valuable advice to improve the quality of this paper. We have tried our best to improve and made some changes in the manuscript. The yellow part that has been revised according to your comments.

Comments 1: Table 2. Primers 3, 5, 7 ,12, 14, 16, 18, 20, 32, 34, 36 ,45 are described in the table as "Ferverse primer..." I think these should be described in the table as "Forward primer..."

Response 1: Thank you for pointing them out, and we agree this is an excellent suggestion. Therefore, we have replaced Ferverse with Forward (Table 2 from line 121 to line 122).

Comments 2: Primer 10 in Table 2 is described as "Dorward primer..." I think this should be "Reverse primer..".

Response 2: We sincerely thank you for careful reading. According to your nice suggestion, we have replaced Dorward with Reverse (Primer 10 in Table 2, Line 121).

We appreciate for your warm work earnestly, and thank you very much for your comments and suggestion.

Reviewer 2 Report

The authors did not respond to reviewer's concerns, they just stated that some further experiments will be performed in separate study. Still, important controls are missing, as the expression of heterologous genes have not been determined.

The authors improved some parts of the text.

Author Response

Major comments: The authors did not respond to reviewer's concerns, they just stated that some further experiments will be performed in separate study. Still, important controls are missing, as the expression of heterologous genes have not been determined.

Response : Thank you for your decision and constructive comments on our manuscript. We have carefully considered the suggestion of you, however, we did not include a negative control (HJ, HJ transformed with empty vector p426 and so on) for several reasons. As we said before, we explored the possibility of resveratrol production from glucose as sole carbon source through genome integration of resveratrol biosynthetic pathway using the rice wine yeast S. cerevisiae HJ . Firstly, there are many evidents that it will be not the synthesis of resveratrol in yeast which has not been transferred to the relevant genes (Figure 1). Secondly, resveratrol was detected according to the fermentation results of HJ01-HJ08 through HPLC detection as many references methods, indicating that heterologous genes had been expressed in HJ. In addition, we have also verified the expression of related genes in HJ strains by Polymerase chain reaction (PCR). Last but not least, we have carefully read a large number of literatures according your comments, and in these reports, there were no relevant negative controls (13,15,16) as well as the determination of the expression of heterologous genes. 

  We appreciate for your warm work earnestly, and thank you very much for your comments and suggestion.

Round 3

Reviewer 2 Report

The authors responded to revewer's concerns.

The authors responded to revewer's concerns.